# Comprehensive Analyses of SARS-CoV-2 Transmission in a Public Health Virology Laboratory

**DOI:** 10.3390/v12080854

**Published:** 2020-08-05

**Authors:** Neta S. Zuckerman, Rakefet Pando, Efrat Bucris, Yaron Drori, Yaniv Lustig, Oran Erster, Orna Mor, Ella Mendelson, Michal Mandelboim

**Affiliations:** 1Central Virology Laboratory, Ministry of Health, Chaim Sheba Medical Center, Tel-Hashomer, P.O. Box 5265601, Ramat-Gan, Israel; Neta.Zuckerman@sheba.health.gov.il (N.S.Z.); Rakefet.Pando@sheba.health.gov.il (R.P.); Efrat.Bucris@sheba.health.gov.il (E.B.); Yaron.Drori@sheba.health.gov.il (Y.D.); yaniv.lustig@sheba.health.gov.il (Y.L.); Oran.Erster@sheba.health.gov.il (O.E.); orna.mor@sheba.health.gov.il (O.M.); Ella.Mendelson@sheba.health.gov.il (E.M.); 2The Israel Center for Disease Control, Israel Ministry of Health, P.O. Box 5265601, Tel-Hashomer, Israel; 3Department of Epidemiology and Preventive Medicine, School of Public Health, Sackler Faculty of Medicine, Tel-Aviv University, P.O. Box 39040, Tel-Aviv, Israel

**Keywords:** 2019-nCoV, SARS-CoV-2, COVID-19, staff, infection, next generation sequencing (NGS)

## Abstract

SARS-CoV-2 has become a major global concern as of December 2019, particularly affecting healthcare workers. As person-to-person transmission is airborne, crowded closed spaces have high potential for rapid virus spread, especially early in the pandemic when social distancing and mask wearing were not mandatory. This retrospective study thoroughly investigates a small-scale SARS-CoV-2 outbreak in Israel’s central virology laboratory (ICVL) in mid-March 2020, in which six staff members and two related family members were infected. Suspicions regarding infection by contaminated surfaces in ICVL facilities were nullified by SARS-CoV-2 negative real time polymerase chain reaction (PCR) of work surfaces swipe tests. Complete SARS-CoV-2 genomes were sequenced and mutation analyses showed inclusion of all samples to clades 20B and 20C, possessing the spike mutation D614G. Phylogenetic analysis clarified transmission events, confirming S1 as having infected at least three other staff members and refuting the association of a staff member’s infected spouse with the ICVL transmission cluster. Finally, serology tests exhibited IgG and IgA antibodies in all infected individuals and revealed the occurrence of asymptomatic infections in additional staff members. This study demonstrates the advantages of molecular epidemiology in elucidating transmission events and exemplifies the importance of good laboratory practice, distancing and mask wearing in preventing SARS-CoV-2 spread, specifically in healthcare facilities.

## 1. Introduction

SARS-CoV-2 (severe acute respiratory syndrome coronavirus 2) is a novel coronavirus that emerged in Wuhan, China in December 2019 [1] and has rapidly spread across China and to many countries worldwide, causing severe respiratory disease leading to substantial morbidity and mortality [2,3,4,5,6]. This novel virus is a potential threat to human health worldwide and a major global health concern due to person-to-person transmission and current lack of vaccination and effective therapeutic options [3,7]. Major SARS-CoV-2 worldwide clades have been proposed by nomenclature systems including Nextstrain [8] and the global initiative on sharing all influenza data (GISAID, https://www.gisaid.org) [9]. These are based on viral genomes from >57,000 sequences submitted in GISAID [9]. For example, using Nextstrain’s nomenclature, there are currently five major clades: 19A (the root clade) and 19B, and clades 20A, B and C that are widespread in Europe and include a mutation in the spike protein, D614G, that is associated with increased infectivity and higher viral loads [10].

Non-SARS-CoV-2 human coronaviruses have been circulating worldwide since the late 1960s [11,12]. The current rate of circulation of SARS-CoV-2 in Israel in the winter season is still unknown; however, analysis of Israeli samples during 2015–2016 winter season revealed that they circulate simultaneously with other common respiratory viruses, with 10% human coronavirus positive cases [13].

SARS-CoV-2 circulation in the general population in Israel and worldwide is being assessed using real time PCR (RT-PCR). A rapid development of RT-PCR diagnostic tests specific for SARS-CoV-2 genes has enabled fast and accurate laboratory tests for suspected individuals [14]. These tests were successfully evaluated in Israel’s central virology laboratory (ICVL), where SARS-CoV-2 suspected samples were exclusively examined starting from the first importation case of SARS-CoV-2 into Israel at the end of February until the middle of March 2020. Starting with the first suspected sample in Israel, all samples received in ICVL facilities were dealt with using the strictest safety directions and BSL2+ safety conditions [15,16]. Until mid-March 2020, all SARS-CoV-2 positive cases in Israel were isolated in a designated quarantine facility; however, social distancing and mandatory mask-wearing were not customary or enforced at that time in Israel.

In mid-March 2020, several cases of SARS-CoV-2 infection were identified in ICVL, some of which probably originated from an infected worker, as speculated by the inquiry-based epidemiological investigation. SARS-CoV-2 airborne transmission was demonstrated to be the most efficient among all transmission routes [17,18] and contagious even in the pre-symptomatic stages [19,20], such that silent virus spread easily occurs. Infection at workplaces was shown as a common transmission route in Israel in the early stages of the SARS-CoV-2 spread, probably facilitated, in the case of the ICVL outbreak, by crowded workspaces and lack of social distancing and mask wearing at that time.

This study thoroughly investigates the local SARS-CoV-2 ICVL outbreak by examining infected ICVL workers, several epidemiologically-related family members, and work surfaces from ICVL facilities. Application of SARS-CoV-2 whole genome next generation sequencing (NGS), RT-PCR, serology tests and phylogenetic tree analyses elucidate person-to-person transmission events, map individual and common mutations and examine suspicions regarding contaminated surfaces. This study demonstrates the added value of molecular epidemiology based on complete viral genomes in elucidating person-to-person transmission, reveals silent infections in non-symptomatic ICVL staff members via serology testing and confirms that the strict safety regulations observed in ICVL most likely prevented further spread of the virus.

## 2. Materials and Methods

### 2.1. Sample Collection, Nucleic Acid Extraction and Viral Genome Quantification by Real-Time PCR (q-PCR)

Immediately following the identification of the first ICVL infection case (S1) on March 15th 2020, nasopharyngeal swabs from all 56 ICVL staff members and another ten non-ICVL staff who worked at the lab around this time were collected, most of them on the same day and a few on the next day. This comprehensive screening test was performed only once. Additional tests for ICVL staff were conducted for a symptomatic individual (*n* = 1), symptomatic relatives (*n* = 2) or for essential workers who were required to work at the laboratory (*n* = 2). Viral genomes were extracted from 200 µL respiratory samples with the MagNA PURE 96 (Roche, Mannheim, Germany), according to the manufacturer instructions and qRT-PCR reactions using primers corresponding to the SARS-CoV-2 envelope (E) gene were performed as previously described [14]. All samples were tested for the human RNAseP gene, which served as a housekeeping gene. The Quantitative reverse transcription PCR (qRT-PCR) reactions were performed in 25 µL Ambion Ag-Path Master Mix (Life Technologies, Carlsbad, CA, USA) using TaqMan Chemistry on the ABI 7500 instrument. Nucleic extraction samples from SARS-CoV-2 positive staff members (S1–S6) and related family members (S7—S4′s spouse and S8—S3′s spouse) were taken for further molecular analysis.

### 2.2. Specific Amplification of SARS-CoV-2 from Clinical Samples

RNA in extracted nucleic acids was reverse transcribed to single strand cDNA using SuperScript IV (ThermoFisher Scientific, Waltham, MA, USA) as per manufacturer’s instructions. SARS-CoV-2 specific primers designed to capture SARS-CoV-2 whole genome (version 1—total 218 primers, divided into two primer pools designed by Josh Quick from ARTIC Network) were used to generate double strand cDNA and amplify it via PCR using Q5 Hot Start DNA Polymerase (NEB) [21]. Briefly, each sample underwent two PCR reactions with primer pool 1 or 2 and 5X Q5 reaction buffer, 19 mM dNTPs and nuclease-free water. Resulting DNA was combined and quantified with Qubit dsDNA BR Assay kit (ThermoFisher Scientific) as per manufacturer’s instructions and 1ng of amplicon DNA in 5 µL per sample was taken into library preparation.

### 2.3. Library Preparation and Sequencing

Libraries were prepared using NexteraXT library preparation kit and NexteraXT index kit V2 as per manufacturer’s instructions (Illumina, San Diego, CA, USA). Libraries were purified with AMPure XP magnetic beads (Beckman Coulter, Brea, CA, USA) and library concentration was measured by Qubit dsDNA HS Assay kit (Thermo Fisher Scientific, Waltham, MA, USA). Library validation and mean fragment size was determined by TapeStation 4200 via DNA HS D1000 kit (Agilent, Santa Clara, CA, USA). The mean fragment size was ~400 bp, as expected. The library mean fragment size and concentration molarity was calculated and each library was diluted to 4 nM. Libraries were pooled, denatured and diluted to 10pM and sequenced on MiSeq with V3 2X300 bp run kit (Illumina). Sequences are available in GISAID (accession numbers: EPI_ISL_435284, EPI_ISL_435286, EPI_ISL435287, EPI_ISL435289, EPI_ISL435291, EPI_ISL_435292, EPI_ISL447250, EPI_ISL447251).

### 2.4. Bioinformatics Analyses

Fastq files were subjected to quality control using FastQC (www.bioinformatics.babraham.ac.uk/projects/fastqc/) and MultiQC [22] and low-quality sequences were filtered using trimmomatic [23]. To obtain a consensus sequence per sample, paired-end fastq files were combined for each sample via Unix cat command. SARS-CoV-2 reference genome was downloaded from the national center for biotechnology information (NCBI) (NC_045512.2) and indexed using Burrows-Wheeler aligner (BWA) [24]. Combined fastq files were mapped to the indexed reference genome using BWA mem [24]. SAMtools suite [25] was used to convert sam to bam files, remove duplicates and filter unmapped reads. Bam files were sorted, indexed and subjected to quality control using SAMtools suite. Coverage and depth of sequencing was calculated from sorted bam files using a custom perl script. Integrative genome viewer (IGV) was used to observe sequencing coverage per position along the genome [26]. A consensus sequence was constructed for each sample using SAMtools mpileup and bcf tools [27] and converted to a fasta file using seqtk (https://github.com/lh3/seqtk).

Multiple alignment of the sequences with the NC_045512.2 reference and additional sequences was carried out using clustal omega [28].

For the phylogenetic tree construction, the general time reversible model with proportion of invariable sites and gamma plus invariant site-distributed rate heterogeneity (GTR + G + I model) was chosen using jModelTest 2 [29]. The phylogenetic tree was constructed and visualized via MEGA7 [30] using the maximum likelihood method with 1000 bootstrap runs.

Clade annotations were attained from Nextstrain [8], who identify variants that define clades of interest from sequences submitted to GISAID by labs worldwide and updates them periodically in clades.tsv (https://github.com/nextstrain/ncov/blob/master/defaults/clades.tsv).

Additional bioinformatic analyses such as translation from nucleotide to amino acid sequences, comparison of differences across sequences and sample clustering were conducted and visualized using R and Bioconductor packages Seqinr [31], HDMD (https://CRAN.R-project.org/package=HDMD) and ggplot2 [32]. Classification to amino acid groups was set according to physiochemical attributes determined by Atchley et al. [33].

### 2.5. Wipe Test Sampling

Immediately following the identification of the first ICVL infection case (S1), six samples (as detailed in Table 1) were obtained from all SARS-CoV-2 relevant surfaces and equipment in the ICVL using into ∑-Virocult (Medical Wire, Corsham, UK) virus transport medium as described [34]. Three of the samples were taken from the biosafety cabinets (BSCs) in which all suspected SARS-CoV-2 were opened, while the other three samples were obtained from all other surfaces in these facilities including door knobs, the outer surface of all equipment in the room, etc., with special attention to “high-touched areas”. All samples were tested for SARS-CoV-2 using qRT-PCR as mentioned above.

### 2.6. Serology

IgG and IgA antibodies against SARS-CoV-2 were detected by an in-house ELISA using antigen prepared as described [35]. For the ELISA, a 96 well microtiter Polysorb plate (Nunc, Thermo, Roskilde, Denmark) was coated overnight at 4 °C with 1 µg/mL of RBD antigen for detection of IgG and 2 µg/mL for detection of IgA antibodies. After blocking with 5% skimmed milk at 25 °C for 60 min, human serum samples (diluted 1:100 with 3% skimmed milk), were added to antigen coated wells. The plate was incubated at 25 °C for 120 min, washed and goat anti-human IgG horseradish peroxidase (HRP) conjugate (Jackson ImmunoResearch, Philadelphia, PA, USA) (diluted 1:15,000) or anti human IgA HRP conjugate (Abcam, Cambridge, MA, USA) was added to each well for 60 min. After addition of TMB substrate and stop solution (1 M HCl) the OD of each well was measured at 450 nm. ELISA index value below 0.9 was considered negative, between 0.9 and 1.1, equivocal and equal, and above 1.1, positive, respectively. In a validation study which included 633 serum samples obtained from 309 persons infected by SARS-CoV-2 and 324 healthy, uninfected individuals, specificity and sensitivity of the IgG was 98% and 88%, respectively and specificity and sensitivity of the IgA was 98% and 80%, respectively.

## 3. Results

### 3.1. SARS-CoV-2 Specific qRT PCR Assay Identifies SARS-CoV-2 Positive ICVL Staff and Relatives

Following confirmed SARS-CoV-2 in ICVL staff member S1, SARS-CoV-2 specific qRT-PCR assays were conducted for all ICVL staff and relevant family members on March 15, 2020 and on the next day. Only two additional staff members tested positive for SARS-CoV-2 (Table 2)—S2, with a relatively high cycle threshold (Ct) (Ct = 33.07) and S3 with a much lower Ct (Ct = 18.77). A few days later, on March 23rd, two additional ICVL staff members S4 and S6 tested positive for SARS-CoV-2 (Ct = 26 and 22 respectively). On March 29th, an additional qRT-PCR assay for all self-isolated ICVL staff was conducted as a precaution test prior to their return to the lab. Surprisingly, an additional ICVL staff member—S5, tested positive (Ct = 28.58) albeit non-symptomatic.

The spouse of S3 (S8) presented with symptoms, post-exposure to a verified SARS-CoV-2 individual, and was therefore examined and found positive on March 18th (Ct = 22). The spouse of S4 (S7) also presented with symptoms and was found positive on March 29th (Ct = 24). All infected individuals were in an age range of 39–65 and were non-symptomatic or exhibited mild symptoms (Table 2). An inquiry-based epidemiological investigation was conducted for each infected individual (data not shown) and estimated dates of infection were inferred. Definitive information regarding transmission was not attainable for each worker or related family member. Nevertheless, within the ICVL transmission chain, patient S1 was determined as having infected multiple individuals according to the epidemiological investigation.

### 3.2. Whole Genome Sequencing-Based Molecular Epidemiology Elucidates Transmission Events

To molecularly infer transmission within the ICVL local outbreak, SARS-CoV-2 was amplified in all SARS-CoV-2 positive samples from ICVL staff and related family members using specific primers, and whole genomes were sequenced from each sample via Illumina technology. All samples had >99% coverage and an average sequencing depth of 13,583, with the exception of S2 which had 98% coverage, probably due to a high cycle threshold (Ct) value (i.e., lower virus quantity) (Table 2).

A phylogenetic tree was constructed to depict relationships amongst all samples. All samples sequenced were associated with clade 20, harboring the clade’s mutations A23403G and C14408T (Nextstrain nomenclature [8]). All ICVL staff members exhibited the 20C clade-defining mutations G25563T and C1059T. S8 (S3′s spouse) lacked one of the 20C mutations C1059T; Interestingly, S7 (S4′s spouse), exhibited three additional mutations, G28881A, G28882A and G28883C, which are associated with clade 20B. S1, S4 and S5 had identical sequences (Figure 1), while S6 exhibited only one difference—a uniquely observed mutation in nsp12—G15243T (Figure 1 and Table 3). Given that an inquiry-based epidemiological investigation determined S1′s date of infection to an earlier time compared to S4, S5 and S6 (Table 2), these results confirm that S1 infected at least three other ICVL workers (S4, S5 and S6). Two additional ICVL members, S2 and S3, exhibited two and one differences, respectively, compared to the identical S1, S4 and S5 sequences (Table 3), and additional mixed-nucleotide differences characteristic of a quasi-species (Appendix A). Although they shared the 20C clade-defining mutation C1059T with ICVL members S1, S4 and S5, which produced a high-confidence split from non-ICVL members S7 and S8 (bootstrap value of 73%, Figure 1), given their additional mutations, which may have been acquired individually or from a different transmission chain, S2 and S3 cannot be placed within the ICVL transmission chain. S8 was placed in the 20C clade due to a shared mutation with all ICVL members, G25563T, in addition to its own unique mutation. However, S7 was placed outside of the ICVL transmission chain due to two mutations from a related but different clade, 20B. These conclusions are further supported by a larger-scale phylogenetic tree including the ICVL workers and family member samples in addition to 25 randomly chosen Israel-based samples downloaded from GISAID, wherein S7 is placed with non-ICVL samples with 88% confidence (Appendix A).

### 3.3. Mutations along SARS-CoV-2 whole Genome within a Transmission Chain

To assess mutational positions along the genome, ICVL-related sequences were aligned to the SARS-CoV-2 reference genome (NC_045512.2) and mutations along the whole genome were enumerated and their impact was assessed. Overall, 14 mutations were observed in the 5′UTR, nsp2, nsp3, nsp12, nsp13, nsp14, spike, orf3a and nucleocapsid regions in the SARS-CoV-2 genome (Figure 2 and Table 3). As specified, 7/14 mutations were associated with the 20B and 20C clades (Table 3), 4/14 mutations were detected in all sequences compared to the reference sequence (C241T, C3037T, C14408T, A23403G), and the rest of the mutations (3/14) were uniquely observed in S2, S3, S6, S7 and S8 (Table 3). 10/13 of mutations in translated regions led to a replacement (R) in the amino acid and 8/13 were R mutations that also to a change in the amino acid (AA) classification group, which might possess a greater conformational or structural impact on the resulting protein. 3/13 mutations were silent (S), i.e., did not lead to a change in the amino acid, and these were mostly translated to an amino acid within the aliphatic AA group. Additional changes that were observed uniquely in S2 and S3 sequences included mixtures of nucleotides most likely resulting from quasi-species (Appendix A). Overall, these results demonstrate active mutation accumulation in various regions along the genome—some mutations were uniquely observed in some individuals, while others were clade-associated mutations which were also observed in thousands of samples worldwide, attesting to their stability following person-to-person transmission.

### 3.4. Wipe Test for the Detection of SARS-CoV-2 on Laboratory Surfaces

In order to exclude surface infection origin, we sampled the relevant surfaces in the ICVL facility and tested them for the SARS-CoV-2 via qRT-PCR (Table 1). All BSCs were sampled separately from the rest of the surfaces, as a SARS-CoV-2 positive BSC in the inner surface does not necessarily indicate the BSC as an infection origin to the lab worker. All of the examined surfaces tested negative for SARS-CoV-2, including highly touched areas out of the BSC, verifying that they were not involved in the infection. Despite having found all surfaces negative for SARS-CoV-2, the whole ICVL facility was thoroughly decontaminated immediately after shut-down as a measure of precaution.

### 3.5. Serological Analysis of all ICVL Staff Members and Relatives

Serological analysis of sera was conducted for all 56 ICVL staff members and infected spouses (S7 and S8) immediately after return to the lab. Analysis results clearly indicate that all verified SARS-CoV-2 positive staff members and their spouses (S1–S8) had high levels of both SARS-CoV-2 IgG and IgA antibodies. Surprisingly, three additional ICVL staff members also tested positive for both SARS-CoV-2 IgG and IgA, although with lower levels compared to the verified SARS-CoV-2 individuals (data not shown) and three additional staff members had intermediate and positive levels of IgA only (data not shown). All other ICVL staff tested negative for SARS-COV-2 IgG and IgA. Overall IgG and IgA positivity was 14.2% and 16% respectively.

## 4. Discussion

Since December 2019, the spread of SARS-CoV-2 has become a major global concern. Small-scale but meaningful outbreaks led to the shutdown of numerous laboratories, hospital wards and clinics worldwide, at the time that they were most needed. In the case of the ICVL outbreak, all relevant staff were tested for SARS-COV-2 and sent for 14 days home isolation immediately after the identification of the first infected worker, as defined by Israel’s ministry of health based on Center for Disease Control and Prevention (CDC) recommendations [36,37,38]. An additional test was performed upon return to the lab from quarantine. During the 14-day isolation period, three additional ICVL staff members tested positive for SARS-COV-2, one of whom was identified at the 14th day of isolation though having tested negative two days earlier. The median incubation period of SARS-CoV-2 has been reported to be five days [2,39]. The quarantine for exposed individuals is set to 14 days as there are few cases (~1%) detected later, while after 14 days it is highly improbable that further symptomatic infections would be detected [39,40].

Epidemiological investigations, specifically in healthcare settings, are essential for review and correction of regulations in order to avoid similar future situations. In the case of the ICVL outbreak, the questionnaire-based epidemiological investigation was not able to track the exact whereabouts of each infected individual in the lab or determine exact person-to-person contacts. As a precaution, all ICVL staff were sent for a 14-day home-isolation and the laboratory was shut down, disabling a great deal of SARS-CoV-2 testing throughout Israel in mid-March 2020, when SARS-CoV-2 was starting to spread in the country. In this study, the usage of molecular epidemiology was able to clarify the transmission events, i.e., confirm the existence of S1 as having infected at least three other ICVL workers (S4, S5 and S6). Two additional ICVL workers, S2 and S3, were placed in the same clade as the S1, S4, S5 and S6 cluster. However, the numerous additional unique mutations they exhibited may have been acquired via internal virus replication or indicate relation to a different transmission chain not sampled in this study. The analysis also determined that S7, the spouse of ICVL worker S4, was not infected by S4 and was probably associated with a different transmission chain, as evidenced by the three mutations associated with a different clade found in S7 only, and the high confidence split from rest of the ICVL transmission chain. S8 (S3′s spouse), shared only one 20C clade mutation with other ICVL members—G25563T, while lacking the other—C1059T. Given that, according to the epidemiological investigation, S8 was exposed to an additional SARS-CoV-2 verified patient aside from S3, Thus, S8 cannot assuredly be placed within the ICVL transmission chain. Alternatively, S8 may have contracted the virus from the two different sources and carried a mixed viral population.

A thorough analysis of the different mutations accumulated by each sample demonstrated a limited number of mutations in some individuals and numerous mutations in others, in several locations/genes along the SARS-CoV-2 genome. Lack of or a limited number of mutations is evidence of transmission, as demonstrated by the identical sequences of the S1–S4–S5 cluster and S6, with one difference compared to these. Mutations which were associated with worldwide clades 20B and 20C suggest that they have survived selection events over time and have a better chance of being transmitted. Indeed, the A23403G mutation associated with clade 20, also referred to as D614G (numbering according to the position of the mutation in the spike protein amino acid sequence), was recently shown to have emerged in Europe and is now the most prevalent form worldwide [10]. Other mutations were uniquely observed in individual samples and are representative of the viral population in the individual at the time the sample was taken. A closer examination of the nature of these mutations shows that two of the unique mutations were silent, i.e., did not lead to a change in the amino acid, while others were replacement mutations that led to a change in the amino acid, and in some instances also the attribute group of amino acids, which may lead to a change in the resulting protein. Depending upon whether these mutations are beneficial for the virus, they may survive and be transmitted over time, for example, as occurred with the emerging clade 20-associated mutation A23403G/D614G that changed aspartate (acidic group) to glycine (aliphatic group) in the spike protein which presumably provides better viral fitness [10]. Because of the nature of each mutation and lack of additional known samples related to this transmission chain, it is difficult to determine whether a mutation uniquely observed in an individual is part of a different transmission chain that is not represented in the tree, or a momentary mutation observed only at the time of sampling that will not survive over time.

Another issue arising in outbreaks in healthcare facilities is whether transmission occurred by an infectious individual or from infected surfaces. Viral agent viability on surfaces was found to be a for a few hours up to a few days [41]. Specifically, SARS-CoV-2 was reported to survive on surfaces for up to nine days, depending on the type of surface [42]. Aerosols spread from clinical samples are most significant in clinical hospital laboratories due to aerosol formation during the tube uncapping stage. Despite SARS-CoV-2′s relatively high surface survival, its lipid envelope renders it susceptible to all commonly used disinfectants generally used in clinical laboratories [42,43]. Hence, good laboratory disinfection practice is sufficient to prevent SARS-CoV-2 laboratory-acquired infection. Indeed, to date, laboratory-acquired infection has not been reported for SARS-CoV-2 [44]. In ICVL too, suspicions regarding contaminated surfaces were nullified by SARS-CoV-2 negative qRT-PCR of all laboratory surfaces where samples were handled, thus confirming a person-to-person transmission in this local outbreak.

In light of emerging evidence of SARS-CoV-2′s silent spread [45,46,47], serology tests for all ICVL staff were conducted to examine the extent of infection in the ICVL facility. All verified infected SARS-CoV-2 staff and related family members were IgG and IgA positive, having been tested at least two weeks following the last negative SARS-CoV-2 result. These results are in accordance with latest publications demonstrating that IgG antibodies for SARS-CoV-2 appear in 79.8% of the patients at days 10–15 after onset of disease, and within 19 days after symptom onset 100% of the tested patients were positive for IgG [48,49]. Surprisingly, six additional ICVL staff members tested positive for one or both IgG/IgA antibodies, four of whom reside in an area having had high SARS-CoV-2 incidence at that time and were infected, despite being non-symptomatic, supporting the later appearance of low IgG antibody titers. In spite of a thorough questionnaire-based epidemiological investigation, the origin of this local ICVL outbreak remains unclear, although the molecular epidemiological investigation based on whole genome SARS-CoV-2 sequencing sheds light on ICVL person-to-person transmission events. In addition, suspicions regarding contaminated surfaces were nullified by SARS-CoV-2 negative qRT-PCR of all laboratory surfaces where samples were handled, and it is now clear that the nature of infection was by person-to-person transmission. Since then, lessons regarding transmission of SARS-CoV-2 have been learned and applied both in Israel and worldwide, where social distancing and mask-wearing are now mandatory and specifically enforced in healthcare facilities.

## Figures and Tables

**Figure 1 viruses-12-00854-f001:**
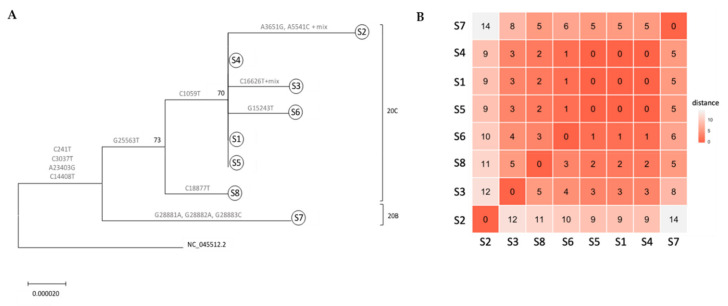
Whole genome-based phylogenetic tree of ICVL local outbreak. (**A**) Phylogenetic tree of six ICVL samples (S1–S6), two family-related samples (S7–S8) and SARS-CoV-2 reference sequence (NC 045512.2). Tree was inferred by maximum likelihood based on the GTR + I + G evolutionary model. The robustness of branching pattern was tested by 1000 bootstrap replications and the percentage of successful bootstrap replicates is indicated at the nodes, where only values of >70% are shown. Mutational positions separating the samples are shown by each branch and corresponding 20B and 20C clades (according to Nextstrain nomenclature) are indicated. (**B**) Sample similarity clustering, calculated according to hamming distance (the number of different nucleotides across the whole genome sequences calculated for each pair of samples). Red represents the least difference and white the most. The number of differences is also noted.

**Figure 2 viruses-12-00854-f002:**
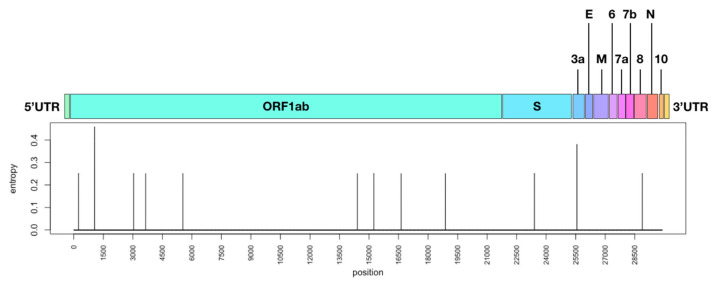
Diversity of mutations along the SARS-CoV-2 whole genome. All mutations observed along the genome and their frequency, as determined by normalized entropy, where entropy = 0 represents an invariable site and entropy = 1 represents a site where all states occur with equal probability. UTR—untranslated region, ORF—open reading frame, S—spike, E—envelope, M—membrane, N—nucleocapsid.

**Table 1 viruses-12-00854-t001:** Wipe test sampling plan.

Sample No.	Facility	Sampled Equipment
1	Specimen reception area	All work surfaces and equipment in the room including knobs, chairs, doors, etc.
2	Specimen reception area	Biosafety cabinets’ (BSC) outside and inside surface
3	Specimen sampling room #1	All work surfaces and equipment in the room including knobs, chairs, doors, etc.
4	Specimen sampling room #1	BSC outside and inside surface
5	Specimen sampling room #2	All work surfaces and equipment in the room including knobs, chairs, doors, etc.
6	Specimen sampling room #2	BSC outside and inside surface

**Table 2 viruses-12-00854-t002:** Infected cases characteristics and sequencing parameters.

	Cases Characteristics						Sequencing Parameters		
Sample No.	SARS-CoV-2 Ct	IgG/IgA Values	Age	Date of Detection	Estimated Date of Infection*	Transmission Chain	# Mapped Reads	% Coverage	Avg. Depth
**S1**	14.3	3.36/3.34	55	15.3.20	unknown	ICVL	3,806,897	100.00	6095
**S2**	33.07	4.87/13.86	65	15.3.20	unknown	NA	5,096,580	98.00	5357
**S3**	18.77	4.52/5.9	46	15.3.20	10.3.20	NA	3,495,706	99.65	5501
**S4**	26	4.77/6.77	39	23.3.20	14.3.20	ICVL	1,739,690	99.24	4713
**S5**	28.58	4.71/1.83	61	29.3.20	14.3.20	ICVL	1,419,044	99.96	4958
**S6**	22	4.75/1.76	41	23.3.20	14.3.20	ICVL	3,380,868	99.90	6014
**S7**	24	6.69/2.1	41	29.3.20	14.3.20	NA	8,580,675	99.99	40,466
**S8**	22	7.17/10.23	52	18.3.20	10.3.20	NA	9,498,576	100	45,041

Characteristics for each case include SARS-CoV-2 cycle threshold (Ct), age, date of detection and estimated date of infection according to the epidemiological investigation and association with the Israel Central Virology Laboratory (ICVL) transmission chain. Sequencing characteristics include the total number of sequence reads mapped to SARS-CoV-2 reference (# mapped reads), percent of SARS-CoV-2 genome covered by the sequence reads (% coverage) and the average depth of sequencing per sample. * Estimated date of infection according to the epidemiological investigation

**Table 3 viruses-12-00854-t003:** Specific nucleotide and AA mutations in all infected cases.

Gene	Clade	Nuc #	REF	S1	S2	S3	S4	S5	S6	S7	S8	AA #	REF	S1	S2	S3	S4	S5	S6	S7	S8	R/S	AA Group
**5’UTR**		241	c	**t**	**t**	**t**	**t**	**t**	**t**	**t**	**t**	-	-	-	-	-	-	-	-	-	-	-	-
**NSP2**	20C	1059	c	**t**	**t**	**t**	**t**	**t**	**t**	c	c	85	T	**I**	**I**	**I**	**I**	**I**	**I**	T	T	R	hydroxilated (T), aliphatic (I)
**NSP3**		3037	c	**t**	**t**	**t**	**t**	**t**	**t**	**t**	**t**	107	F	F	F	F	F	F	F	F	F	S	aromatic (F)
		3651	a	a	**g**	a	a	a	a	a	a	311	Q	Q	**R**	Q	Q	Q	Q	Q	Q	R	aminic (Q), basic (R)
		5541	a	a	c	a	a	a	a	a	a	941	Q	Q	**P**	Q	Q	Q	Q	Q	Q	R	aminic (Q), proline (P)
**NSP12**	20B	14408	c	**t**	**t**	**t**	**t**	**t**	**t**	**t**	**t**	314	P	L	L	L	L	L	L	L	L	R	proline (P), aliphatic (L)
		15243	g	g	g	g	g	g	t	g	g	601	C	C	C	C	C	C	**F**	C	C	R	cysteine (C), aromatic (F)
**NSP13**		16626	c	c	c	**t**	c	c	c	c	c	131	L	L	L	L	L	L	L	L	L	S	aliphatic (L)
**NSP14**		18887	c	c	c	c	c	c	c	c	**t**	280	L	L	L	L	L	L	L	L	L	S	aliphatic (L)
**SPIKE**	20B	23403	**a**	**g**	**g**	**g**	**g**	**g**	**g**	**g**	**g**	614	D	**G**	**G**	**G**	**G**	**G**	**G**	**G**	**G**	R	acidic (D), aliphatic (G)
**ORF3a**	20C	25563	g	**t**	**t**	**t**	**t**	**t**	**t**	g	**t**	58	Q	**H**	**H**	**H**	**H**	**H**	**H**	Q	**H**	R	basic (H), aminic (Q)
**NUCAP**	20B	28881	g	g	g	g	g	g	g	**a**	g	203	R	R	R	R	R	R	R	**K**	R	R	basic (R/K)
	20B	28882	g	g	g	g	g	g	g	**a**	g	204	R	R	R	R	R	R	R	**K**	R	R	basic (R/K)
	20B	28883	g	g	g	g	g	g	g	**c**	g	204	G	G	G	G	G	G	G	R	G	R	aliphatic (G), basic (R)

Mutated positions are shown for each sample (bolded and highlighted), including association with the gene (gene) and known clade-defining mutations (clade), the position of the mutation in the nucleotide or AA sequence (nuc/AA), the nucleotides or AA in the position for the samples and reference sequence (REF), whether they led to a replacement or a silent mutation (R/S), and the AA group.

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
