# Peer review of "Comprehensive Analyses of SARS-CoV-2 Transmission in a Public Health Virology Laboratory"

_viruses, 2020, doi:10.3390/v12080854_

Round 1

Reviewer 1 Report

This paper documents the person to person transmission of SARS-CoV-2 within a public health virology laboratory that received clinical samples to be tested for SARS-CoV-2.  The main scientific advantage offered by the results of this study is a very limited but controlled and documented evidence of person to person transmission within the confined spaces of a public health laboratory.  The circumstances that resulted in the infection of the laboratory workers was documented from the first or index case through the other individuals and family that had contact with the initial case.  Since this occurred in a public health laboratory, appropriate sampling and testing could be done in a controlled experiment. While the consensus is that SARS-CoV2 is transmitted via aerosols, there are few studies that provide data to support this concept.

The study is interesting in that it provides information on the transmission of the coronavirus within public health laboratories that were working under BSL2 protocols and not wearing facial coverings (masks) to protect them from airborne transmission of the virus.  We are certainly much further along in our understanding of the transmission of SARS 2 than at the time this study was done.  So, the question is whether this study provides any new evidence on the epidemiology and ecology of this coronavirus infection.  The concept of aerosol transmission of SARS 2 is now the dogma for this virus, so this study provides no new information that the virus is transmitted via aerosols, but it does provide a limited experimental documentation of this route of transmission.

The authors have provided an adequate experimental design and done very well in documenting the timeline of transmission of the coronavirus among 6 laboratory workers and 2 family members that also presented with symptoms and were tested.  The sequence and phylogenetic analyses support that s1 was the index case and transmitted virus to colleagues s2-6.  Viral sequences and the phylogenetic tree verify that the viral genetics of S1-6 are essentially the same sequence with some minor mutations.  The one family member s8 seems to have been infected with the same virus as S1-6 but s7 has sufficient sequence variation to suggest that this family member was infected through another source and not the laboratory worker.

Authors the sequencing data is pertinent to your showing and explaining the transmission of the virus within the confines of the public health laboratory.  In your discussion, lines 290 to 311, on the mutations suffice it so say that there are some variations but do not speculate as to what these mutations may mean related to the biology of the virus, you have no data to support this.  It is OK to make suggestions as to what these mutations may indicate or if they may indicate a pattern of mutations.  Please review this rather lengthy paragraph and shorten to what is pertinent to your study. 

Authors, it is good that you sampled various surfaces to determine if the SARS 2 virus could have been transmitted to laboratory workers through contaminated workplaces.  However, it is not clear from your description in Materials and Methods if this was one series of wipe tests at one period or whether wipe tests were taken over a period?  Please explain.  Also, there is the question on when during the working day the wipe tests were done.  Authors if these samples were always taken after disinfection of the laboratory surfaces, then PCR negative  results would be expected.  However, this would seem to lead to the erroneous conclusion that surfaces are not a source of the virus. If you took samples from surfaces after processing samples prior to disinfecting the surfaces perhaps virus was present on these surfaces for a limited period.   You do not indicate when and how often you did your wipe tests, so please explain when this was done for clarification.  Also, line 250 in the paper indicates that these results are in Table 2, did not see a column listing the wipe tests of the surfaces?  Either correct this or indicate the data in Table 2.

Authors did you quantitate the level or titer of IgG and IgA or was the ELISA as described in the Materials and Methods a qualitative determination.  You indicate that all infected persons had antibodies, but you do not present any data to indicate this.  Why not at least indicate in Table 2, the level of the ELISA results to provide data to validate your results?  Did the three additional staff members that tested positive for both IgG and IgA antibodies to SARS-CoV-2 have contact with s1-s6?  It would be interesting to know if this was the case and if not then perhaps, they were infected from another source.

In lines 333-336 you indicate that the IgA antibodies were present in naïve individuals.  How do you know these individuals were not infected with the virus hence the antibody response and if you sampled after you found they had antibody then evidence of the virus may have  been eliminated?  An IgA response would not need to come from digestion of contaminated food, you would expect an IgA response with a respiratory virus.  Also, there is no evidence of food borne transmission of this virus.  You have no evidence for this, and other evidence would refute transmission through food.  Likely, these individuals had enough contact with s1-6 to allow for airborne transmission.  The discussion of IgA and food transmission of the virus should be eliminated. 

Other concerns: 

Line 243 says Table 1, should this be Table 3?

Line 177, how did you compute your average sequence depth?  You have a wide range of average depth from 4,713 to 45,041.  Is it possible that the large depth results for s7 and s8 may skew you average, as s1-s6 are 10-fold lower?  Please explain and perhaps expressing this a two average depths s1-s6 and s7 and s8?  Perhaps this makes no difference but consider the question.

Title could perhaps be shortened to “Comprehensive analyses of the transmission of SARS-CoV2 outbreak in a public health virology laboratory.”  Your main point is the transmission of this virus in a defined space, more that the molecular analysis.  The molecular analysis supports your hypothesis of the transmission of this virus between laboratory workers in a SARS-CoV-2 clinical processing area.

This paper documents the person to person transmission of SARS-CoV-2 within a public health virology laboratory that received clinical samples to be tested for SARS-CoV-2.  The main scientific advantage offered by the results of this study is a very limited but controlled and documented evidence of person to person transmission within the confined spaces of a public health laboratory.  The circumstances that resulted in the infection of the laboratory workers was documented from the first or index case through the other individuals and family that had contact with the initial case.  Since this occurred in a public health laboratory, appropriate sampling and testing could be done in a controlled experiment. While the consensus is that SARS-CoV2 is transmitted via aerosols, there are few studies that provide data to support this concept.

The study is interesting in that it provides information on the transmission of the coronavirus within public health laboratories that were working under BSL2 protocols and not wearing facial coverings (masks) to protect them from airborne transmission of the virus.  We are certainly much further along in our understanding of the transmission of SARS 2 than at the time this study was done.  So, the question is whether this study provides any new evidence on the epidemiology and ecology of this coronavirus infection.  The concept of aerosol transmission of SARS 2 is now the dogma for this virus, so this study provides no new information that the virus is transmitted via aerosols, but it does provide a limited experimental documentation of this route of transmission.

The authors have provided an adequate experimental design and done very well in documenting the timeline of transmission of the coronavirus among 6 laboratory workers and 2 family members that also presented with symptoms and were tested.  The sequence and phylogenetic analyses support that s1 was the index case and transmitted virus to colleagues s2-6.  Viral sequences and the phylogenetic tree verify that the viral genetics of S1-6 are essentially the same sequence with some minor mutations.  The one family member s8 seems to have been infected with the same virus as S1-6 but s7 has sufficient sequence variation to suggest that this family member was infected through another source and not the laboratory worker.

Authors the sequencing data is pertinent to your showing and explaining the transmission of the virus within the confines of the public health laboratory.  In your discussion, lines 290 to 311, on the mutations suffice it so say that there are some variations but do not speculate as to what these mutations may mean related to the biology of the virus, you have no data to support this.  It is OK to make suggestions as to what these mutations may indicate or if they may indicate a pattern of mutations.  Please review this rather lengthy paragraph and shorten to what is pertinent to your study. 

Authors, it is good that you sampled various surfaces to determine if the SARS 2 virus could have been transmitted to laboratory workers through contaminated workplaces.  However, it is not clear from your description in Materials and Methods if this was one series of wipe tests at one period or whether wipe tests were taken over a period?  Please explain.  Also, there is the question on when during the working day the wipe tests were done.  Authors if these samples were always taken after disinfection of the laboratory surfaces, then PCR negative  results would be expected.  However, this would seem to lead to the erroneous conclusion that surfaces are not a source of the virus. If you took samples from surfaces after processing samples prior to disinfecting the surfaces perhaps virus was present on these surfaces for a limited period.   You do not indicate when and how often you did your wipe tests, so please explain when this was done for clarification.  Also, line 250 in the paper indicates that these results are in Table 2, did not see a column listing the wipe tests of the surfaces?  Either correct this or indicate the data in Table 2.

Authors did you quantitate the level or titer of IgG and IgA or was the ELISA as described in the Materials and Methods a qualitative determination.  You indicate that all infected persons had antibodies, but you do not present any data to indicate this.  Why not at least indicate in Table 2, the level of the ELISA results to provide data to validate your results?  Did the three additional staff members that tested positive for both IgG and IgA antibodies to SARS-CoV-2 have contact with s1-s6?  It would be interesting to know if this was the case and if not then perhaps, they were infected from another source.

In lines 333-336 you indicate that the IgA antibodies were present in naïve individuals.  How do you know these individuals were not infected with the virus hence the antibody response and if you sampled after you found they had antibody then evidence of the virus may have  been eliminated?  An IgA response would not need to come from digestion of contaminated food, you would expect an IgA response with a respiratory virus.  Also, there is no evidence of food borne transmission of this virus.  You have no evidence for this, and other evidence would refute transmission through food.  Likely, these individuals had enough contact with s1-6 to allow for airborne transmission.  The discussion of IgA and food transmission of the virus should be eliminated. 

Other concerns: 

Line 243 says Table 1, should this be Table 3?

Line 177, how did you compute your average sequence depth?  You have a wide range of average depth from 4,713 to 45,041.  Is it possible that the large depth results for s7 and s8 may skew you average, as s1-s6 are 10-fold lower?  Please explain and perhaps expressing this a two average depths s1-s6 and s7 and s8?  Perhaps this makes no difference but consider the question.

Title could perhaps be shortened to “Comprehensive analyses of the transmission of SARS-CoV2 outbreak in a public health virology laboratory.”  Your main point is the transmission of this virus in a defined space, more that the molecular analysis.  The molecular analysis supports your hypothesis of the transmission of this virus between laboratory workers in a SARS-CoV-2 clinical processing area.

This paper documents the person to person transmission of SARS-CoV-2 within a public health virology laboratory that received clinical samples to be tested for SARS-CoV-2.  The main scientific advantage offered by the results of this study is a very limited but controlled and documented evidence of person to person transmission within the confined spaces of a public health laboratory.  The circumstances that resulted in the infection of the laboratory workers was documented from the first or index case through the other individuals and family that had contact with the initial case.  Since this occurred in a public health laboratory, appropriate sampling and testing could be done in a controlled experiment. While the consensus is that SARS-CoV2 is transmitted via aerosols, there are few studies that provide data to support this concept.

The study is interesting in that it provides information on the transmission of the coronavirus within public health laboratories that were working under BSL2 protocols and not wearing facial coverings (masks) to protect them from airborne transmission of the virus.  We are certainly much further along in our understanding of the transmission of SARS 2 than at the time this study was done.  So, the question is whether this study provides any new evidence on the epidemiology and ecology of this coronavirus infection.  The concept of aerosol transmission of SARS 2 is now the dogma for this virus, so this study provides no new information that the virus is transmitted via aerosols, but it does provide a limited experimental documentation of this route of transmission.

The authors have provided an adequate experimental design and done very well in documenting the timeline of transmission of the coronavirus among 6 laboratory workers and 2 family members that also presented with symptoms and were tested.  The sequence and phylogenetic analyses support that s1 was the index case and transmitted virus to colleagues s2-6.  Viral sequences and the phylogenetic tree verify that the viral genetics of S1-6 are essentially the same sequence with some minor mutations.  The one family member s8 seems to have been infected with the same virus as S1-6 but s7 has sufficient sequence variation to suggest that this family member was infected through another source and not the laboratory worker.

Authors the sequencing data is pertinent to your showing and explaining the transmission of the virus within the confines of the public health laboratory.  In your discussion, lines 290 to 311, on the mutations suffice it so say that there are some variations but do not speculate as to what these mutations may mean related to the biology of the virus, you have no data to support this.  It is OK to make suggestions as to what these mutations may indicate or if they may indicate a pattern of mutations.  Please review this rather lengthy paragraph and shorten to what is pertinent to your study. 

Authors, it is good that you sampled various surfaces to determine if the SARS 2 virus could have been transmitted to laboratory workers through contaminated workplaces.  However, it is not clear from your description in Materials and Methods if this was one series of wipe tests at one period or whether wipe tests were taken over a period?  Please explain.  Also, there is the question on when during the working day the wipe tests were done.  Authors if these samples were always taken after disinfection of the laboratory surfaces, then PCR negative  results would be expected.  However, this would seem to lead to the erroneous conclusion that surfaces are not a source of the virus. If you took samples from surfaces after processing samples prior to disinfecting the surfaces perhaps virus was present on these surfaces for a limited period.   You do not indicate when and how often you did your wipe tests, so please explain when this was done for clarification.  Also, line 250 in the paper indicates that these results are in Table 2, did not see a column listing the wipe tests of the surfaces?  Either correct this or indicate the data in Table 2.

Authors did you quantitate the level or titer of IgG and IgA or was the ELISA as described in the Materials and Methods a qualitative determination.  You indicate that all infected persons had antibodies, but you do not present any data to indicate this.  Why not at least indicate in Table 2, the level of the ELISA results to provide data to validate your results?  Did the three additional staff members that tested positive for both IgG and IgA antibodies to SARS-CoV-2 have contact with s1-s6?  It would be interesting to know if this was the case and if not then perhaps, they were infected from another source.

In lines 333-336 you indicate that the IgA antibodies were present in naïve individuals.  How do you know these individuals were not infected with the virus hence the antibody response and if you sampled after you found they had antibody then evidence of the virus may have  been eliminated?  An IgA response would not need to come from digestion of contaminated food, you would expect an IgA response with a respiratory virus.  Also, there is no evidence of food borne transmission of this virus.  You have no evidence for this, and other evidence would refute transmission through food.  Likely, these individuals had enough contact with s1-6 to allow for airborne transmission.  The discussion of IgA and food transmission of the virus should be eliminated. 

Other concerns: 

Line 243 says Table 1, should this be Table 3?

Line 177, how did you compute your average sequence depth?  You have a wide range of average depth from 4,713 to 45,041.  Is it possible that the large depth results for s7 and s8 may skew you average, as s1-s6 are 10-fold lower?  Please explain and perhaps expressing this a two average depths s1-s6 and s7 and s8?  Perhaps this makes no difference but consider the question.

Title could perhaps be shortened to “Comprehensive analyses of the transmission of SARS-CoV2 outbreak in a public health virology laboratory.”  Your main point is the transmission of this virus in a defined space, more that the molecular analysis.  The molecular analysis supports your hypothesis of the transmission of this virus between laboratory workers in a SARS-CoV-2 clinical processing area.

Author Response

Comment: This paper documents the person to person transmission of SARS-CoV-2 within a public health virology laboratory that received clinical samples to be tested for SARS-CoV-2.  The main scientific advantage offered by the results of this study is a very limited but controlled and documented evidence of person to person transmission within the confined spaces of a public health laboratory.  The circumstances that resulted in the infection of the laboratory workers was documented from the first or index case through the other individuals and family that had contact with the initial case.  Since this occurred in a public health laboratory, appropriate sampling and testing could be done in a controlled experiment. While the consensus is that SARS-CoV2 is transmitted via aerosols, there are few studies that provide data to support this concept.

The study is interesting in that it provides information on the transmission of the coronavirus within public health laboratories that were working under BSL2 protocols and not wearing facial coverings (masks) to protect them from airborne transmission of the virus.  We are certainly much further along in our understanding of the transmission of SARS 2 than at the time this study was done.  So, the question is whether this study provides any new evidence on the epidemiology and ecology of this coronavirus infection.  The concept of aerosol transmission of SARS 2 is now the dogma for this virus, so this study provides no new information that the virus is transmitted via aerosols, but it does provide a limited experimental documentation of this route of transmission.

The authors have provided an adequate experimental design and done very well in documenting the timeline of transmission of the coronavirus among 6 laboratory workers and 2 family members that also presented with symptoms and were tested.  The sequence and phylogenetic analyses support that s1 was the index case and transmitted virus to colleagues s2-6.  Viral sequences and the phylogenetic tree verify that the viral genetics of S1-6 are essentially the same sequence with some minor mutations.  The one family member s8 seems to have been infected with the same virus as S1-6 but s7 has sufficient sequence variation to suggest that this family member was infected through another source and not the laboratory worker.

Authors the sequencing data is pertinent to your showing and explaining the transmission of the virus within the confines of the public health laboratory.  In your discussion, lines 290 to 311, on the mutations suffice it so say that there are some variations but do not speculate as to what these mutations may mean related to the biology of the virus, you have no data to support this.  It is OK to make suggestions as to what these mutations may indicate or if they may indicate a pattern of mutations.  Please review this rather lengthy paragraph and shorten to what is pertinent to your study. 

Response: Thank you for this comment, we have altered this paragraph as suggested (lines 328-332).

Authors, it is good that you sampled various surfaces to determine if the SARS 2 virus could have been transmitted to laboratory workers through contaminated workplaces.

Response: We thank the reviewer for his comment.  Please find below our point-by-point response to the reviewer concerns.

Comment: However, it is not clear from your description in Materials and Methods if this was one series of wipe tests at one period or whether wipe tests were taken over a period?  Please explain.

Response: Thank you for this comment. The swipe test sampling was only conducted immediately following the identification of the first ICVL infection case (S1). This was clarified at line 148 in the text.

Comment: Also, there is the question on when during the working day the wipe tests were done.  Authors if these samples were always taken after disinfection of the laboratory surfaces, then PCR negative results would be expected.

Response: The swipe tests were taken once, on the evening of March 15th, immediately after the detection of S1 as SARS-CoV-2 positive. We added a new table, Table 1, describing the sampling plan of the tests. Disinfecting agents cause a reduction in viral agents, to very low, sometimes detectable levels. Therefore, it was highly important to perform the swipe tests to all of the working surfaces in the BSCs, but also to all other equipment located in the working area including "non-work" surfaces, making sure they were not contaminated with SARS-CoV-2.

Comment: However, this would seem to lead to the erroneous conclusion that surfaces are not a source of the virus. If you took samples from surfaces after processing samples prior to disinfecting the surfaces perhaps virus was present on these surfaces for a limited period. You do not indicate when and how often you did your wipe tests, so please explain when this was done for clarification. 

Response: As mentioned, the swipe test were taken once, on the evening of March 15th, immediately after the detection of S1 as SARS-CoV-2 positive. Indeed, the tests were taken after the disinfecting of the relevant work spaces. However, our main concern was that SARS-CoV-2 may be present on other "non-work” spaces". Therefore, we sampled the whole room including all furniture, equipment, doors etc. as mentioned in the added Table 1.

Comment: Also, line 250 in the paper indicates that these results are in Table 2, did not see a column listing the wipe tests of the surfaces?  Either correct this or indicate the data in Table 2.

Response: Indeed, these data do not appear in Table 2, we referred to the relevant Table 1.

Comment: Authors did you quantitate the level or titer of IgG and IgA or was the ELISA as described in the Materials and Methods a qualitative determination. 

Response: The ELISA is a qualitative assay. As described in the Materials and Methods, the ELISA’s output is an index value. However, because it is qualitative there is a cut-off which differentiates between positive and negative results. In addition, values that are 0.1 below or above the cut-off are considered borderline (or equivocal). Naturally, a higher index value usually indicates higher antibody levels, however because the assay is not linear it is not quantitative.

Comment: You indicate that all infected persons had antibodies, but you do not present any data to indicate this.  Why not at least indicate in Table 2, the level of the ELISA results to provide data to validate your results? 

Response: In light of the reviewers comment, serological qualitative results and their index value were added to Table 2.

Comment: Did the three additional staff members that tested positive for both IgG and IgA antibodies to SARS-CoV-2 have contact with s1-s6?  It would be interesting to know if this was the case and if not then perhaps, they were infected from another source.

Response: Thank you for this comment. Indeed, these workers are part of the ICVL staff, resulting in ordinary work relations with S1-S6. None of these workers were in exceptionally closer contact with S1-S6. However, four of them reside in an area having had high SARS-CoV-2 incidence at that time, suggesting they may have been infected outside of the ICVL.

Comment: In lines 333-336 you indicate that the IgA antibodies were present in naïve individuals.  How do you know these individuals were not infected with the virus hence the antibody response and if you sampled after you found they had antibody then evidence of the virus may have been eliminated?  An IgA response would not need to come from digestion of contaminated food, you would expect an IgA response with a respiratory virus. 

Response: We completely agree and this is only a hypothesis due to the fact that serum IgA is different than mucosal IgA and since no IgG was detected it suggest a difference in the infection pathway. 

Comment: Also, there is no evidence of food borne transmission of this virus.  You have no evidence for this, and other evidence would refute transmission through food.  Likely, these individuals had enough contact with s1-6 to allow for airborne transmission.  The discussion of IgA and food transmission of the virus should be eliminated. 

Response: As mentioned above, this was only a hypothesis raised in the discussion, however in light of the reviewer’s comment we removed this sentence (lines 368-371) from the manuscript

Other concerns: 

Comment: Line 243 says Table 1, should this be Table 3?

Response: Thank you for this comment, this typo was corrected to Supplementary table 1 (Table S1).

Comment: Line 177, how did you compute your average sequence depth?  You have a wide range of average depth from 4,713 to 45,041.  Is it possible that the large depth results for s7 and s8 may skew you average, as s1-s6 are 10-fold lower?  Please explain and perhaps expressing this a two average depths s1-s6 and s7 and s8?  Perhaps this makes no difference but consider the question.

Response: The depth of sequencing in each position in the genome was recorded, and minimum, average and maximum depth was calculated across all positions in the genome for each individual separately. The large differences in average depths across samples are a result of differences in the total number of reads allocated for each sample for sequencing.

Comment: Title could perhaps be shortened to “Comprehensive analyses of the transmission of SARS-CoV2 outbreak in a public health virology laboratory.”  Your main point is the transmission of this virus in a defined space, more that the molecular analysis.  The molecular analysis supports your hypothesis of the transmission of this virus between laboratory workers in a SARS-CoV-2 clinical processing area.

Response: We thank the reviewer for this comment, we shortened the title as the reviewer suggested.

Reviewer 2 Report

Zuckerman et al. investigated a SARS-CoV-2 outbreak at a SARS diagnostic lab in Israel. The study is well conducted and multiple analyses were performed, including virus molecular characterization, serology, and search of the source of infection. The manuscript is well written, the methods are valid, and the conclusions are mostly supported by their data. However, I think the manuscript lacks details in the M&M section and some clarity in the results section (see below for details). Finally, I have some comments about data interpretation that I think authors should address before I recommend this paper for publications.

- Lines 168-9 and 285-6. How do you know for sure that it was S1 who infected the other subjects when S1, S2 and S3 were diagnosed during the same day?

 - The tree in Figure S2, which I suggest including as main figure, does not support S8 as being part of the ICVL outbreak. In fact, S8 sequence is not included in the same clade as all other strains (although bootstraps are low). It is possible that, like for S7, the source of infection was different. I suggest rebuilding this tree with additional sequences to try and clarify this aspect and give more power to the bootstrapping analysis. Similarly, you should not exclude entirely the possibility that other subjects (e.g. S2 that is fairly different from other strains) also acquired the infection from a different source.

- Introduction. To give a proper background, you should discuss viral classification, including clades nomenclature and distribution.

- Section 2.1. The timeline for sample collection and testing in the methods is not clear and should be specified further. How many testing sessions were performed and in which dates? Also, how was SARS-CoV-2 infection diagnosed?  Furthermore, it is not clear how many samples and how many individuals were tested. Initially you tested 66, but afterwards not all subjects were re-screened. Can you provide the number of how many individuals were re-screened? Also, how many relatives were screened or re-screened? 

- Line 82. What kind of samples were collected? This is relevant for RNA quantification.

- Line 92. You should provide a reference for these primers. Also, which protocol version was used?

- Lines 122-5. There is something weird about the phylogenetic analysis method section. I think you meant “maximum likelihood” and not “maximum composite likelihood”? There are also some discrepancies about the numbers of bootstraps between this section and figure S2 caption.

- Sequences should be submitted to GenBank and accession numbers provided.

- Section 2.5. This is a bit confusing and it is not clear how many samples were tested. Were they 6 in total or 6 for each surface? Can you provide a list of tested surfaces (even as supplementary material)?

- Section 2.6. What is the specificity of this ELISA? Is it possible that the ELISA detected antibodies against seasonal CoVs in some of the other subjects that were positive in serology but had negative swabs?

- Lines 154-7. Please check this paragraph because it’s confusing: S2 is mentioned twice (both as original and follow-up detection) and the timeline is weird since S1, S2, and S3 were all discovered the same day.

- Lines 179-185. This part also needs to become clearer. Furthermore, C14408T and A23403G are common to both 20B and 20C. Also, what about C3037T and G28883C? These two clade-identifying mutations are reported in the tree in Figure 1 and in the Table but are not mentioned in the text.

-Do you think that the lack of C1059T in S8 is a reversion or that S8 represents an intermediate strain?

- Table 1 can be made clearer with some footnotes (e.g. explain what do the columns “clade” and “R/S” stand for).

Minor:

- Line 47. Since human coronaviruses are common respiratory viruses I’d change “with common respiratory viruses” with “with other common respiratory viruses”.

- Line 47. “Human” should not be capitalized.

- Line 78. “and other 10 non-ICVL staff”.

- Line 80. “Samples from SARS-CoV-2 positive staff”.

- Line 95. “Qubit”

- Line 250: these results are not reported in Table 2…

- Line 251. Did you mean S1-S8?

- Section 3.5. What was the overall seroprevalence (for IgA, IgG and both)?

Author Response

Comment: Zuckerman et al. investigated a SARS-CoV-2 outbreak at a SARS diagnostic lab in Israel. The study is well conducted and multiple analyses were performed, including virus molecular characterization, serology, and search of the source of infection. The manuscript is well written, the methods are valid, and the conclusions are mostly supported by their data. However, I think the manuscript lacks details in the M&M section and some clarity in the results section (see below for details). Finally, I have some comments about data interpretation that I think authors should address before I recommend this paper for publications.

Response: We thank the reviewer for the comments.  Please find below our point-by-point response to the reviewer’s concerns.

Comment: Lines 168-9 and 285-6. How do you know for sure that it was S1 who infected the other subjects when S1, S2 and S3 were diagnosed during the same day?

Response: Thank you for this comment. We determined that S1 infected S4, S5 and S6 according to both epidemiological investigation and the identical SARS-CoV-2 sequences. This was clarified in the text in lines 189-190. In addition, as replied to your next comment, we agree that S2 and S3 could not assuredly be placed in the ICVL transmission chain and therefore S1 had more indications of having infected S4, S5 and S6. This was also clarified in the text in Table 2 and in lines 215-219.

Comment: The tree in Figure S2, which I suggest including as main figure, does not support S8 as being part of the ICVL outbreak. In fact, S8 sequence is not included in the same clade as all other strains (although bootstraps are low). It is possible that, like for S7, the source of infection was different. I suggest rebuilding this tree with additional sequences to try and clarify this aspect and give more power to the bootstrapping analysis. Similarly, you should not exclude entirely the possibility that other subjects (e.g. S2 that is fairly different from other strains) also acquired the infection from a different source.

Response: We agree with the reviewer that S8 cannot assuredly be placed within the ICVL transmission chain, given that S8 shares the 20C clade-defining mutation G25563T with the other ICVL members but lacks another (C1059C). We have now clarified this point in the discussion (line 312-316). Similarly, we agree with the reviewer that S2 may be related to a different transmission chain, given its many unique mutations, and have clarified this point as well, both in the results section (line 215-219) and the discussion (line 306-309).

Comment: Introduction. To give a proper background, you should discuss viral classification, including clades nomenclature and distribution.

Response: The relevant background regarding the clades was added to the introduction (line 43-48).

Comment: Section 2.1. The timeline for sample collection and testing in the methods is not clear and should be specified further. How many testing sessions were performed and in which dates? Also, how was SARS-CoV-2 infection diagnosed?  Furthermore, it is not clear how many samples and how many individuals were tested. Initially you tested 66, but afterwards not all subjects were re-screened. Can you provide the number of how many individuals were re-screened? Also, how many relatives were screened or re-screened?

Response: Thank you for this comment. This section was rephrased and data were added to section 2.1 as the reviewer suggested (lines 82-95).

Comment: Line 82. What kind of samples were collected? This is relevant for RNA quantification.

Response: Thank you for this comment. The nasopharyngeal swabs nucleic extraction were taken for further analysis. This was clarified in line 82 in the text.

Comment: Line 92. You should provide a reference for these primers. Also, which protocol version was used?

Response: Thank you for comment. As stated in the methods section, the primers used were from the ARTIC group. We have now added a link to the protocol (line 102).

Comment: Lines 122-5. There is something weird about the phylogenetic analysis method section. I think you meant “maximum likelihood” and not “maximum composite likelihood”? There are also some discrepancies about the numbers of bootstraps between this section and figure S2 caption.

Response: The relevant text in the methods section was corrected according to the reviewer’s comments (lines 138 and Figure S2 caption).

Comment: Sequences should be submitted to GenBank and accession numbers provided.

Response: Sequences are already submitted to GISAID, accession numbers were added to the methods section (line 115).

Comment: Section 2.5. This is a bit confusing and it is not clear how many samples were tested. Were they 6 in total or 6 for each surface? Can you provide a list of tested surfaces (even as supplementary material)?

Response: Thank you for this comment, indeed we examined 6 samples, however, each sample was taken from several work spaces at designated SARS-CoV-2 work spaces: the Specimen reception area and two Specimen sampling rooms. As the reviewer suggested, we added Table 1 to section 2.5.

Comment: Section 2.6. What is the specificity of this ELISA? Is it possible that the ELISA detected antibodies against seasonal CoVs in some of the other subjects that were positive in serology but had negative swabs?

Response: In a validation study which included 633 serum samples obtained from 309 individuals infected by SARS-CoV-2 and 324 of healthy, uninfected individuals, specificity and sensitivity of the IgG was 98% and 88% respectively and specificity and sensitivity of the IgA was 98% and 80%, respectively. This was added to the Materials and Methods section.

Due to the large cohort which was used to test specificity and the high percentage of individuals infected yearly with seasonal CoVs cross-reactivity is very unlikely.

Comment: Lines 154-7. Please check this paragraph because it’s confusing: S2 is mentioned twice (both as original and follow-up detection) and the timeline is weird since S1, S2, and S3 were all discovered the same day.

Response: The relevant text was corrected in the manuscript (lines 174-176).

Comment: Lines 179-185. This part also needs to become clearer. Furthermore, C14408T and A23403G are common to both 20B and 20C. Also, what about C3037T and G28883C? These two clade-identifying mutations are reported in the tree in Figure 1 and in the Table but are not mentioned in the text.

Response: The relevant paragraph has been clarified and corrected to include all clade-defining mutations appearing in the tree (lines 204-208). Regarding C3037 and C241T - they are associated with clade 20, however they’re not currently clade-defining (according to Nextstrain definitions: https://github.com/nextstrain/ncov/blob/master/defaults/clades.tsv)

Comment: Do you think that the lack of C1059T in S8 is a reversion or that S8 represents an intermediate strain?

Response: An examination of these mutations in Nextstrain’s SARS-CoV-2 global analysis (https://nextstrain.org/ncov/global) shows that S8 is not unique and that there are many samples globally that have G25563T but not C1059T, and thus they remain “outside” of the 20C clade (and classified as 20A). It is possible that they will spontaneously develop the C1059T mutation in the future if it is a position prone to mutations, or on the other hand develop another mutation that will be beneficial for the virus and will eventually form its own clade. A reversion mutation is also hypothetically possible, although one would need to study samples from the same individuals over time to evaluate this possibility, and the mutation would have to not be crucial for virus evolution (survival, spread, etc.) to be reverted easily.

Comment: Table 1 can be made clearer with some footnotes (e.g. explain what do the columns “clade” and “R/S” stand for).

Response: Thank you for this comment. We added a legend to the relevant table with all of the required interpretations. 

Minor:

Comment: Line 47. Since human coronaviruses are common respiratory viruses I’d change “with common respiratory viruses” with “with other common respiratory viruses”.

Response: The sentence was changed as the reviewer suggested.

Comment: Line 47. “Human” should not be capitalized.

Response: The capitalization was removed.

Comment: Line 78. “and other 10 non-ICVL staff”.

Response: The word "and" was added.

Comment: Line 80. “Samples from SARS-CoV-2 positive staff”.

Response: The words "Samples from" were added.

Comment: Line 95. “Qubit”

Response: The typo was corrected.

Comment: Line 250: these results are not reported in Table 2…

Response: Indeed this data does not appear in table 2, we referred to the relevant Table 1.

Comment: Line 251. Did you mean S1-S8?

Response: Thank you for this comment. We indeed meant that SARS-CoV-2 positive BSC in its inner surface doesn't necessarily indicate the BSC as an infection origin to the lab worker. We corrected the sentence in the text (lines 269-271).

Comment: Section 3.5. What was the overall seroprevalence (for IgA, IgG and both)?

Response: The IgG and IgA seroprevalence was 14.2% and 16% respectively. The overall seroprevalence (either IgA or IgG was 17.8%). This information was added to the results section (line 283). 

Round 2

Reviewer 1 Report

The authors have adequately addressed the concerns of this reviewer.

Reviewer 2 Report

All my concerns have been addressed and I am happy to recommend the publication of this study.